# Advance of SOX Transcription Factors in Hepatocellular Carcinoma: From Role, Tumor Immune Relevance to Targeted Therapy

**DOI:** 10.3390/cancers14051165

**Published:** 2022-02-24

**Authors:** Xiangyuan Luo, Xiaoyu Ji, Meng Xie, Tongyue Zhang, Yijun Wang, Mengyu Sun, Wenjie Huang, Limin Xia

**Affiliations:** 1Department of Gastroenterology, Institute of Liver and Gastrointestinal Diseases, Tongji Hospital of Tongji Medical College, Huazhong University of Science and Technology, Wuhan 430030, China; luoxiangyuan@hust.edu.cn (X.L.); d202081875@hust.edu.cn (X.J.); xiemeng@hust.edu.cn (M.X.); m201976049@hust.edu.cn (T.Z.); m202076163@hust.edu.cn (Y.W.); sunmengyu@stu.ahmu.edu.cn (M.S.); 2Hubei Key Laboratory of Hepato-Pancreato-Biliary Diseases, Tongji Hospital of Tongji Medical College, Huazhong University of Science and Technology, Wuhan 430030, China; huangwenjie@tjh.tjmu.edu.cn; 3Hepatic Surgery Center, Tongji Hospital of Tongji Medical College, Huazhong University of Science and Technology, Wuhan 430030, China

**Keywords:** liver cancer, transcription factor, SOX, tumor immune microenvironment, treatment

## Abstract

**Simple Summary:**

Hepatocellular carcinoma (HCC) is one of the deadliest human health burdens worldwide. However, the molecular mechanism of HCC development is still not fully understood. Sex determining region Y-related high-mobility group box (SOX) transcription factors not only play pivotal roles in cell fate decisions during development but also participate in the initiation and progression of cancer. Given the significance of SOX factors in cancer and their ‘undruggable’ properties, we summarize the role and molecular mechanism of SOX family members in HCC and the regulatory effect of SOX factors in the tumor immune microenvironment (TIME) of various cancers. For the first time, we analyze the association between the levels of SOX factors and that of immune components in HCC, providing clues to the pivotal role of SOX factors in the TIME of HCC. We also discuss the opportunities and challenges of targeting SOX factors for cancer.

**Abstract:**

Sex determining region Y (SRY)-related high-mobility group (HMG) box (SOX) factors belong to an evolutionarily conserved family of transcription factors that play essential roles in cell fate decisions involving numerous developmental processes. In recent years, the significance of SOX factors in the initiation and progression of cancers has been gradually revealed, and they act as potential therapeutic targets for cancer. However, the research involving SOX factors is still preliminary, given that their effects in some leading-edge fields such as tumor immune microenvironment (TIME) remain obscure. More importantly, as a class of ‘undruggable’ molecules, targeting SOX factors still face considerable challenges in achieving clinical translation. Here, we mainly focus on the roles and regulatory mechanisms of SOX family members in hepatocellular carcinoma (HCC), one of the fatal human health burdens worldwide. We then detail the role of SOX members in remodeling TIME and analyze the association between SOX members and immune components in HCC for the first time. In addition, we emphasize several alternative strategies involved in the translational advances of SOX members in cancer. Finally, we discuss the alternative strategies of targeting SOX family for cancer and propose the opportunities and challenges they face based on the current accumulated studies and our understanding.

## 1. Introduction

Transcription factors (TFs) are one of the most pivotal sequence-specific DNA-binding proteins responsible for decoding DNA sequences. In order to accomplish the transcriptional regulation of genes, TFs require at least one DNA-binding domain (DBD) for recognizing and binding specific DNA sites located in promoters or enhancers/silencers of targeted genes and another effector domain for recruiting transcriptional machinery to DNA [1]. To this point, over 2000 TFs divided into various families based on the homologous DBDs have been identified, many of which enable the controlling of some critical biological processes in a way that depends on cell type and development patterning [2]. These TFs, which account for about 8% of all human genes, are essential for maintaining normal physiological processes [3]. Consequently, mutation or dysregulation of many master TFs usually leads to a wide range of diseases, and approximately 19% of human TFs are involved in the process [3]. Among them, the tumor is the most concerned disorder.

The first oncogenic TFs were found in human acute leukemias [4]. The genes encoding TFs produce fusion proteins with functional alteration due to frequent chromosomal translocations such as PLZF-RARα [5]. These abnormal TFs fusion products disrupt the transcriptional networks during the specific periods of hematopoietic cell differentiation and ultimately cause the initiation of cancer [5]. Additionally, a large number of TFs as drivers or suppressors in solid tumors have been identified. For instance, the MYC TFs are among the most commonly dysregulated driver proteins in an overwhelming majority of human cancers [6]. Upregulated MYC promotes the initiation and progression of cancers by controlling cancer cell-intrinsic oncogenic signaling pathways, tumor microenvironments (TME), and immune surveillance [6]. Similarly, many other TFs such as HIFs, FOXO, and STAT3/5 are also the master regulators that determine the growth, metastasis, and other hallmarks of cancers [7,8,9]. Accordingly, plenty of explorations related to TFs are essential to deeply understand the molecular mechanisms of the initiation and progression of cancers and are beneficial in developing novel cancer therapy strategies. Recently, the sex-determining region Y (SRY)-related high-mobility group (HMG) box (SOX) family TFs, which have been described as playing an essential role in a variety of cancers and as potential therapeutic targets for cancers, is of particular interest.

In this review, we first summarize the SOX family in terms of basic structure, classification, and physiological functions. We will focus on the SOX family in hepatocellular carcinoma (HCC), which is one of the deadliest human health burdens worldwide [10], and provide an updated overview of how SOX members are regulated and how they work on the HCC development by controlling the transcription networks. Given that the tumor immune microenvironment (TIME) markedly affects the development and therapy response of tumors, we also detail the role of SOX members in remodeling TIME and analyze the association between SOX members expression and immune components such as immune cells, immune checkpoints, antigen-presenting molecules, chemokines, and chemokine receptors in HCC for the first time. In addition, we emphasize several alternative strategies involved in the translational advances of SOX members in cancer, and finally discuss the opportunities and challenges of targeting the SOX family in cancer based on the current accumulated studies and on our understanding.

## 2. Overview of the SOX Transcription Factors

In 1990, John Gubbay et al. first discovered the SOX family TFs in mammals, given the presence of an amino acid domain characterized by the HMG box of homology with mammalian testis-determining factor SRY [11]. The HMG box, an evolutionarily conserved DNA-binding motif, is the representative signature of the SOX family. This HMG motif comprises 79 amino acids, but only the sequence RPMNAFMVW is conserved for all SOX factors except SRY [12]. This sequence appears to be the most extensive signature that can identify SOX genes [12]. Under long-term selection pressure, this core and ancestral HMG motif gradually changed in sequence but retained the sequence-specific DNA-binding function, coupled with changes in proteins outside the HMG domain, leading to more than 20 distinct SOX proteins in vertebrates [12].

On the basis of the degree of sequence identity of the HMG motif, SOX members are classified into eight subgroups, including subgroup A (SRY), subgroup B1 (SOX1, SOX2, and SOX3), subgroup B2 (SOX14 and SOX21), subgroup C (SOX4, SOX11, and SOX12), subgroup D (SOX5, SOX6, SOX13, and SOX23), subgroup E (SOX8, SOX9, and SOX10), subgroup F (SOX7, SOX17, and SOX18), subgroup G (SOX15), and subgroup H (SOX30) (Figure 1). In general, the members within the same SOX subgroup have more than 80% of sequence identity in the HMG motif and are also consistent in other conserved sequences [13]. Accordingly, SOX proteins in the same subgroup equip with similar biochemical properties and probably perform repetitive functions [13]. For instance, all members of the SOX D subgroup, in addition to being highly consistent in the HMG motif, also have the characteristics of an abundant and conserved leucine zipper motif, which contributes to forming homodimers or heterodimers [14]. Consequently, SOX5 and SOX6 proteins of the SOX D subgroup show overlapping and cooperative functions in chondrogenesis [15]. On the other hand, many SOX proteins from different subgroups play distinct and even opposed functions though all SOX proteins, recognizing and binding to a common DNA consensus motif [16]. Past studies have revealed that SOX proteins can recognize many different DNA binding sites. The possible factors that give rise to this difference include flanking sequences that impact binding affinity, SOX proteins dimerization, and interaction between SOX proteins with other TFs due to their adjacent binding sites [13]. Additionally, whether the SOX protein possesses a transactivation domain (such as SRY, SOX B1, SOX C, SOX E, SOX F, and SOX G) or a transcriptional repressor domain (such as SOX B2) determines the transcriptional direction [13]. The post-translational modifications of SOX proteins also affect their transcriptional activity [17]. All these factors lead to the difference and diversity of SOX protein functions.

SOX proteins that play essential roles in cell fate decisions involving numerous developmental processes, including neural development, skeletal development, testis development, endoderm development, vascular/lymphatic development, and hair follicle development have been sufficiently studied [18]. Stem cells, as a source of tissue and organ formation that support the growth and development of the body, are usually regulated by context-specific TFs to control their self-renewal and differentiation [19]. Accumulated evidence indicates that the SOX proteins participate in controlling stem and progenitor cell fate and tissue regeneration. By interacting with intracellular cofactors and competing with other TFs of alternative lineages, SOX proteins transcriptionally activate self-renewal-related genes and inhibit differentiation-related genes, therefore reprogramming and maintaining stem cell populations [19]. In addition, SOX proteins also serve as pioneer TFs, firstly combining and preparing target genes for timely activation to maintain cell self-renewal once differentiation occurs [20,21]. Consequently, SOX factors are key regulators of cell fate, and their alteration and dysregulation often lead to various diseases, especially cancer.

## 3. SOX Transcription Factors in Hepatocellular Carcinoma

The initiation and progression of HCC are controlled by a complex and huge molecular regulatory network. TFs in this network integrate and process extracellular and intracellular signaling pathways and output the key commands in the nucleus to affect the malignant phenotype of HCC. Currently, SOX factors, as pivotal regulators in both tumor driver and suppressor signaling pathways, have been reported to control the proliferation, migration, angiogenesis, cancer stem cell (CSCs) properties, and drug resistance of HCC extensively. (Figure 2)

### 3.1. SOX A (SRY)

The *SRY* gene, which locates on the Y chromosome, initiates testicular differentiation and induces masculinization of mammalian embryos. The protein expression of SRY is limited to the pre-Sertoli cells of male mice 10.5–12.5 days after intercourse and is regulated by DNA methylation-mediated epigenetic silencing [22]. Of note, there is evidence that the copy number of the *SRY* gene (Yp11.3) is gained or amplified in 11.8% of male patients with HCC, indicating that SRY may promote HCC development [23]. An early study showed that SRY is highly expressed in HCC cells and tissues and is associated with the poor survival of patients with HCC [24]. However, although SRY is a sex-determining gene, there is no difference in SRY expression between male and female HCC patients. Functionally, SRY promotes the migration and invasion of HCC cells [24]. Simultaneously, S. Murakami et al. reported that SRY promotes the proliferation, invasion, and tumorigenicity of two male mice HCC cell lines by directly activating Sgf29 partly through histone H3 acetylation [25]. Their further study extended to human HCC cell lines and discovered that SRY maintains the CSCs properties, including self-renewal, chemoresistance, and tumorigenicity, by transactivating the classic CSCs marker OCT4 and up-regulating other CSCs markers such as CD13, thereby contributing to HCC progression [26]. In 2017, the role of SRY in hepatocarcinogenesis was verified in vivo for the first time [27]. This research team found that SRY is overexpressed in approximately 84% of male HCC, while most female HCC tissues show as SRY negative [27]. In their in vivo experiment, the aggravating liver cell injury, increased cell proliferation, and accelerated tumor growth were observed in both male and female hepatocyte-specific SRY overexpressing transgenic mice followed by DEN administration [27]. The promotion of SRY in hepatocarcinogenesis is partly attributed to the activation of the SOX9 and PDGFRα/PI3K/Akt pathway, given that these pathways are involved in promoting inflammation, fibrosis, and development of HCC [27,28,29,30].

Taken together, the *SRY* functions as an oncogene to accelerate hepatocarcinogenesis and facilitate HCC development. However, its role and regulatory mechanisms in HCC, especially the difference between male and female HCC, remains to be further explored in light of its particular role in sex-determination for males.

### 3.2. SOX B1 and SOX B2 (SOX1, SOX2, SOX3, SOX14, and SOX21)

The SOX B group is divided into the SOX B1 subgroup and the SOX B2 subgroup. In general, the members of SOX B1 play a transcriptional activation role on their targets, while those in SOX B2 play a transcriptional repression role on theirs [31]. 

SOX1, SOX2, and SOX3 belong to the SOX B1 subgroup. However, these three members display distinct effects in HCC development. *SOX1* as a tumor suppressor gene has been demonstrated in HCC. By hypermethylation of its promoter, SOX1 is frequently downregulated in HCC [32]. SOX1 significantly inhibits the proliferation and invasion of HCC through competing with TCFs to bind β-catenin, subsequently inactivating the Wnt signaling pathway [33]. Moreover, SOX1 induces cellular senescence in Hep3B cells [33]. 

Conversely, SOX2 and SOX3 play roles in promoting the initiation and progression of HCC. It is well known that a unique subpopulation of cells in HCC is present, whether liver CSCs or tumor-initiating cells (TICs), which share features with normal stem or progenitor cells, including self-renewal and differentiation [34]. These liver CSCs/TICs regenerate all malignant phenotypes of tumors due to their stemness-related capabilities, therefore facilitating the development, recurrence, and therapy resistance of HCC [34]. SOX2, as a transcriptionally regulatory center of self-renewal in embryonic stem cells, plays a core role in reprograming adult somatic cells into a pluripotent stem cell-like state [35,36]. Furthermore, studies showed that SOX2 is highly expressed in HCC, and its overexpression is correlated with the poor survival of patients with HCC [37,38]. Consequently, the cancer-promoting effect performed by SOX2 in HCC is likely to be mainly attributable to its ability to control the stemness of HCC cells. Indeed, well-documented evidence links the role of SOX2 in stemness regulation and HCC promotion. Specifically, a study uncovered that Cyclin G1 up-regulates the expression of SOX2 by activating the Akt/mTOR signaling pathway, which leads to an increased proportion and enhanced tumor-initiating capacity of liver TICs, as well as inducing chemoresistance of HCC cells to sorafenib [39]. Also of interest is that blocking the Akt/mTOR signaling pathway by inhibitors significantly enhances the sensitivity of HCC cells to sorafenib, providing a potential combination therapeutic strategy for HCC [39]. Indeed, drug resistance has severely limited the efficacy of sorafenib. As a tyrosine kinase inhibitor (TKI) that mainly acts by inhibiting the RAS/RAF/MEK/ERK signaling pathway, the main neuronal isoform of RAF, BRAF, and MEK pathways and BRAF mutation are important resistance mechanisms of sorafenib [40]. Therefore, combination therapy strategies to eliminate these key resistance factors are the current priority. The activated Akt/mTOR signaling pathway has been demonstrated to confer sorafenib resistance to HCC through multiple mechanisms, including induction of Warburg shift [41] and increased TIC features [42]. The finding that SOX2-mediated the Akt/mTOR signaling pathway induces stemness and sorafenib resistance in HCC complements the stemness-related molecular mechanisms of sorafenib resistance and further confirms the combination therapy of the Akt/mTOR signaling pathway inhibitor and sorafenib for HCC is feasible. In addition, SOX2 is the critical downstream effector of SIRT1, which enhances the self-renewal ability of CSCs and induces HCC development [43]. Mechanistically, SIRT1 deacetylates the histones and interacts with DNMT3A to demethylate the CpGs in the SOX2 promoter, leading to an upregulation of SOX2 transcription [43]. Moreover, Zhang et al. revealed that TGFβ negatively regulates SOX2 in tumor-initiating hepatocytes [44]. According to their study, SOX2 mediates the suppressive role of TGFβ on HCC development through directly transcriptionally activating lncRNA H19 [44]. As we know, the epithelial-mesenchymal transition (EMT) is a cell-biological program that converts the epithelial characteristics into mesenchymal properties, leading tumor cells to lose adhesion and apical-basal polarity, ultimately promoting the invasion and metastases of the tumor [45]. Studies showed that the EMT process induces CSCs properties in tumor cells, indicating that these two cellular states are closely associated [45]. Consistent with this, SOX2, as a key regulator of liver CSCs, also induces the EMT process in HCC [37,46]. Additionally, SOX2 also involves other mechanisms to contribute to the HCC development, including mediating p38a-ROS-induced suppression of hepatocarcinogenesis and activating the CCAT1/EGFR/miR-222-5p/CYLD signal axis to promote HCC progression [38,47]. 

SOX3 is reported to be significantly overexpressed in HCC tissues and associated with HCC development, recurrence, and poor prognosis of patients with HCC [48]. However, its role and related mechanisms in HCC remain unclear, although it acts as an oncogene in many other cancers [49,50].

### 3.3. SOX C (SOX4, SOX11, and SOX12)

Among the members of the SOX C subgroup, SOX4 is the most studied in HCC. Convincing evidence has shown that SOX4, as the direct transcription target of Smad2 and Smad3 downstream of TGFβ, plays a master role in TGFβ-induced EMT, invasion, and metastasis of tumors [51,52]. In HCC, the SOX4 expression is remarkably increased in tumor tissues, especially in metastatic and recurrent HCC samples [53,54,55,56,57,58,59]. The overexpressed SOX4 is positively associated with higher intratumoral microvessel density, distant metastasis, and poor survival of HCC cases, which provides a potential diagnostic and prognostic marker for HCC [56,57,58,59]. As a crucial metastasis-associated regulator, SOX4 induces the chemotaxis of human umbilical vein endothelial cells, angiogenesis, and tumor growth in HCC by activating CXCL12, therefore facilitating HCC metastasis [58]. Treatment with AMD3100, an antagonist targeting CXCR4 (CXCL12 receptor), suppresses the chemotaxis and tube formation of endothelial cells mediated by overexpressed SOX4 [58]. It was also found that SOX4 is overexpressed in liver TICs and plays an indispensable role in liver TICs self-renewal [56]. Mechanistically, SOX4 directly interacts with lncSox4 and STAT3 in the same region of its promotor, thereby up-regulating SOX4 expression and contributing to the liver TICs self-renewal [56]. In addition, as a partner regulator, SOX4 interacts with p53 and specifically impairs the transcription capability of p53 to its downstream target Bax, thus inhibiting the apoptosis of HCC cells following irradiation [54]. Interestingly, although SOX4 acts as a transcription activator in most cancers, its transcriptional repression role has been found in two HCC cell lines [53]. In the future, it is necessary to clarify the distinct function presented by the inverse transcriptional activity of SOX4 in tumors. Past studies have also mined the upstream modifications of SOX4, especially post-transcriptional regulation mediated by microRNA (miRNA). So far, a variety of miRNAs, including miR-129-2, miR-449 family, miR-130a-3p, miR-363-3p, and miR-138 have been suggested to directly or indirectly repress the SOX4 expression in HCC, therefore affecting HCC progression and metastasis [55,57,60,61,62].

Hepatitis B virus (HBV) infection remains the primary risk factor for the initiation and development of HCC [63]. There is evidence that HBV replication is closely related to SOX4 levels and forms a positive regulatory feedback loop [64]. First, HBV up-regulates the SOX4 expression at multiple levels: HBV induces the MAPK/YY1 signaling pathway and subsequently directly transactivates SOX4; HBV up-regulates SOX4 at the post-transcriptional level through abolishing the inhibition of miR-335, miR-129-2, and miR-203 on SOX4 expression; at the post-translational level, HBsAg, the surface antigen of HBV, blocks the SOX4 polyubiquitin by interacting with SOX4 to protect it from proteasome-mediated degradation [64]. Meanwhile, viral-induced SOX4 overexpression, in turn, enhances HBV replication through the interaction between viral genomic DNA and the HMG domain of SOX4 in HCC cells [64]. However, Shu Shi et al. found that the binding sequence (AACAAAG) of SOX4 on HBV proposed by the study mentioned above is only present in 9.17% of all HBV genotype strains [65]. In addition, the conclusion that SOX4 facilitates HBV replication contradicts the previous finding that SOX4 is overexpressed in HCC, while HBV replication levels are low in HCC [66,67]. Accordingly, they further explored the regulatory correlation between SOX4 and HBV and came to an opposing viewpoint that SOX4 represses most HBV replication through inhibiting HNF4α expression, which does not involve the binding sequence of SOX4 on the HBV genome [65]. These inconsistent findings indicate that the intricate and precise role of SOX4 on HBV replication remains not to be fully elucidated, and further research is needed which will provide insight into the molecular mechanism of hepatocarcinogenesis from the etiology.

Accumulated evidence showed that SOX11 is overexpressed in most cancers, including mantle cell lymphoma, glioma, medulloblastoma, ovarian cancer, and breast cancer [68,69,70,71,72]. The key pro-oncogenic role of SOX11 has also been demonstrated in various cancers [73,74]. However, SOX11 serves as a tumor suppressor to participate in HCC progression. Recent studies suggested that the SOX11 expression level is aberrantly downregulated in HCC tissues relative to paired adjacent noncancerous tissues [75,76]. Mechanistically, SOX11 dampens cell proliferation, induces cell cycle arrest, promotes cell apoptosis, and enhances chemosensitivity in HCC through up-regulating NLK1 expression to inactivate the Wnt/β-catenin signaling pathway [76]. In addition, a study also revealed that SOX11, as the direct downstream target of miR-9-5p, is up-regulated by lncMEG3 and positively mediates the role of LncMEG3 in suppressing HCC growth and promoting HCC cell apoptosis [75]. However, given the little evidence so far, the exact function of SOX11 remains controversial.

SOX12 is a novel marker for liver CSCs and is related to tumor metastasis [77,78]. Our previous study unveiled that SOX12 is evidently up-regulated in HCC tissues, especially in HCC metastasis tissues [79]. The overexpressed SOX12 is associated with tumor encapsulation, microvascular invasion, higher TNM stage, and poor prognosis of HCC cases, suggesting that SOX12 could be an independent risk factor for recurrence and worse survival [79]. SOX12 induces the EMT process by directly up-regulating Twist1 expression and facilitates the invasion and metastasis of HCC through transactivating the Twist1 and FGFBP1 expression [79]. Regarding its upstream regulatory mechanism, we disclosed that SOX12 is directly transcriptionally activated by FoxQ1 and plays an indispensable role in FoxQ1-induced HCC metastasis [79]. Additionally, SOX12 is the direct target of miR-874, miR-744, and miR-296-5p to mediate their effects on suppressing the progression and metastasis of HCC [80,81,82].

### 3.4. SOX D (SOX5, SOX6, SOX13, and SOX23)

According to current studies, the members of SOX D exhibit different roles in HCC. SOX6 functions as a tumor suppressor and mediates various miRNAs to affect HCC progression. For instance, miR-155, miR-96, miR-19a-3p, and miR-376c-3p directly bind and repress the SOX6 expression to regulate key molecules or pathways such as the Wnt/β-catenin pathway, thereby promoting HCC progression [83,84,85]. It is interesting to note that SOX6 is the target of miR-1269. However, the single-nucleotide polymorphism (SNP) rs73239138 in miR-1269 interferes with the specific binding to 3′ untranslated region (3’UTR) of SOX6, thus preventing miR-1269 from inhibiting SOX6 [86]. Furthermore, the SOX6 level in HCC tissues is relatively low, and the SOX6 expression is negatively associated with the tumor stage and the serum AFP level [87]. The low SOX6 expression predicts shorter disease-free survival and overall survival, indicating that SOX6 is a potential prognostic marker for HCC [87].

On the contrary, SOX5 and SOX13 play a cancer-promoting role in HCC. The high-expressed SOX5 and SOX13 are frequently observed in HCC tissues compared with paired nontumor tissues [88,89]. In particular, SOX5 is overexpressed in HCC cases with portal vein tumor thrombosis (PVTT), suggesting that it may be involved in the HCC metastasis [88]. Indeed, a former study demonstrated that SOX5 promotes the migration, invasion, and EMT process of HCC in vitro, which is probably attributable to the up-regulation of Twist1 expression [88]. On the other hand, the overexpressed SOX13 is remarkably associated with poor differentiation, metastasis, recurrence, and worse survival of HCC cases [89]. Generally speaking, polymeric complexes are the frequent form of SOX proteins that play their role on a common target [90,91]. A previous study has shown that dimerization of L-SOX5 and SOX6 are co-expressed with SOX9 and act together on the promoter of the chondrocyte differentiation marker Col2a1 [92]. In HCC, SOX13 enables it to dimerize with its partner SOX5 via the coiled-coiled domain, and functionally cooperate to activate Twist1 transcription, thereby promoting the migration, invasion, and EMT process of HCC [89]. In addition, SOX13 also regulates the stem-like properties of HCC cells [93]. Specifically, SOX13 facilitates the proliferation, self-renewal, tumor-initiating, and chemoresistance of HCC cells through inducing TAZ transcription, given that TAZ is an essential effector of CSC properties [93].

### 3.5. SOX E (SOX8, SOX9, and SOX10)

SOX9, which is expressed in the progenitor or precursor cells of the embryonic liver and pancreas, serves as a marker of stem cells and progenitor cells in the liver and pancreas [94]. In HCC, SOX9 is also a crucial CSC marker, given that SOX9 endows the classic CSCs characteristics to HCC cells, including tumorsphere formation and resistance to sorafenib [95,96,97]. Simultaneously, the molecular mechanisms underlying SOX9 induces CSC properties have also been discovered. Chungang Liu et al. uncovered that SOX9 enables increasing symmetrical cell division and enhancing the stemness of liver CSCs via suppressing the Numb expression, a key Notch signaling antagonist that promotes Notch degradation [29]. Symmetrical and asymmetrical cell division are the two main types of cell division during proliferation and development to maintain the self-renewal of cells [29]. Asymmetrical cell division generates one stem cell and one differentiating cell to sustain stem cell homeostasis, while symmetrical cell division leads to two daughter stem cells to augment the stem cell pool [98]. Evidence has suggested that tilting the balance toward symmetrical cell division remarkably elevates the number of CSCs in various tumors [99,100,101]. Herein, this finding in HCC not only expands the tumor pool where symmetrical cell division increases CSCs but also reveals a novel mechanism related to SOX9-induced CSCs properties in HCC. Additionally, SOX9 maintains the liver CSCs properties by transcriptionally up-regulating FZD7 to activate the Wnt/β-catenin signaling pathway [95]. The TGF-β/Smad signaling is another important pathway involved in SOX9 regulating the CSCs features [102]. A previous study revealed that SOX9 directly transactivates circular RNA Circ-FOXP1 to sponge miR-875-3p and miR-421 and activate their respective downstream targets, including SOX9, therefore forming a positive feedback oncogenic loop and leading to the growth and metastasis of HCC [103]. As a feature of CSCs, SOX9 enhances sorafenib resistance potentially through promoting CSCs phenotypes under hypoxic conditions or activating the Akt/ABCG2 pathway [96,97]. Of note, hypoxia, as a crucial hallmark of HCC, is responsible for sorafenib resistance to HCC. The mechanisms underlying hypoxia-induced HCC cells escape from sorafenib include HIF-mediated metabolic reprogramming [104], regulation of the PI3K/AKT signaling pathway [105], etc. The above-mentioned study on SOX9 further expands the mechanism of hypoxia-induced sorafenib resistance and closely links hypoxia, CSCs, and sorafenib sensitivity.

In the adult liver, the SOX9 expression is mainly observed in the cholangiocytes lining the bile ducts, although low SOX9 expression is also detected in hepatocytes surrounding the ductular structures [94]. Nevertheless, substantial studies demonstrated that SOX9 is overexpressed in HCC tissues and is closely related to poor differentiation, venous invasion, high tumor stage, and worse survival of patients with HCC [29,95,106,107,108]. This phenomenon indicates a presence of aberrant upstream modifications that give rise to the dysregulated SOX9 in HCC. Indeed, various miRNAs, including miR-101, miR-138, miR-1-3p, miR-613, miR-5590-3p, miR-520f-3p, and miR-206, have been verified to target SOX9 directly and inhibit its expression [96,107,109,110,111,112,113]. The PTEN is a classic tumor suppressor, which is lost or decreased in approximately 50% of HCC and is frequently phosphorylated-mediated inactivated in 89% of tumor cases [114,115,116]. Intriguingly, in a model with the *Pten* deletion in Sox9+ cells within the liver, heterogeneous tumors consisting of HCC cells and cholangiocarcinoma cells or bile duct adenoma cells are developed [117]. These tumor cells originate from Sox9+ cells, suggesting that *Pten* deletion induces the transformation of the Sox9+ cells [117]. Moreover, following liver injury induced by a high-fat diet or chemical administration, the number of Sox9+ cells expanded significantly, and the formation of these mixed-lineage tumors was remarkably accelerated [117]. Furthermore, the Wnt/β-catenin signaling pathway is needed to maintain the proliferation, survival, and self-renewal of the *Pten* null SOX9+ cells [117]. Consequently, *Pten* deletion can induce the transformation of SOX9+ cells into CSCs, and liver injury activates the transformed SOX9+ cells to promote proliferation and ultimately boost hepatocarcinogenesis [117]. Besides, the mRNA and protein stability of SOX9 is also regulated in HCC. CD73, as a novel marker for liver CSCs, in addition to activating the AKT signaling pathway to promote the SOX9 transcription by c-MYC activation, also protects SOX9 from proteasome-mediated degradation by GSK3β inhibition, therefore maintaining the liver CSCs properties [118]. Similarly, lncDUXAP9 improves the mRNA stability of SOX9 and up-regulates its expression by binding to the 3’UTR of SOX9, thereby sustaining the stemness of HCC [119].

To our knowledge, over 90% of patients with HCC arise in the setting of chronic liver disease, and cirrhosis or liver fibrosis is the most common risk factor [120]. Hepatic stellate cells (HSCs) are an essential driver of liver fibrosis. Their activation state interplays with various resident cells and tumor cells in the TME via releasing multiple cytokines or profibrotic factors to contribute to the growth and metastasis of HCC [121,122,123]. However, the mechanisms underlying the HSCs that facilitate the growth and invasion of HCC remain elusive. Yu Chen et al. disclosed that the up-regulated SOX9 and INHBB in HCC cells promote the growth and metastasis of HCC through activating HSCs presented in the TME [108]. Mechanistically, SOX9 binds to the enhancer of INHBB and induces its expression, thereby increasing the secretion of activin B from HCC cells into the microenvironment, leading to the activation of peri-tumoral HSCs mediated by activin B/Smad signaling and subsequently promoting liver fibrosis and metastasis of HCC [108]. This finding expands the interaction mechanism of HSCs activation on HCC metastasis and provides a potential therapeutic strategy for targeting the communication between HSCs and tumor cells in the TME.

To date, evidence related to the role of SOX8 and SOX10 in HCC is rare. Only one study suggested that the mRNA levels of SOX8 and SOX10 are increased in HCC tissue compared to adjacent benign tissues and healthy tissues [124]. However, the protein level of SOX8 is negatively associated with that of SOX10 in HCC tissues [124]. Functionally, SOX8 promotes the proliferation of HCC cells and activates the Wnt/β-catenin pathway [124]. From this, further related study is needed.

### 3.6. SOX F (SOX7, SOX17, and SOX18) and SOX H (SOX30)

In general, SOX7 functions as a tumor suppressor in most tumors. However, its expression level varies greatly among different tumor types, suggesting the existence of cancer cell-dependent mechanisms to regulate SOX7 expression [125]. In HCC, SOX7 is significantly down-regulated relative to adjacent non-tumor tissue and is associated with the advanced stage of HCC [126,127]. The low SOX7 expression predicts the poor prognosis of patients with HCC, which serves as an independent prognostic factor for HCC [127]. As a tumor-suppressor factor, SOX7 inhibits HCC cell proliferation and induces cell cycle arrest by decreasing the expression of cyclin D1 and c-myc [126]. Zhiyun Zheng et al. found that SOX7 is the target of miR-452 [128]. According to their report, miR-452 maintains the stemness of HCC by targeting SOX7 and directly binding to the β-catenin/TCF/LEF transcriptional factor complex to relieve the suppression of SOX7 to the Wnt/β-catenin signaling pathway [128]. Moreover, miR-184 and miR-935 have also been identified to target SOX7 directly and inhibit its expression [129,130]. Nevertheless, the understanding of the upstream regulatory mechanism of SOX7 remains obscure so far.

In concordance with the SOX17 in colorectal cancer [131], in HCC, SOX17 is frequently methylated at its promoter region and occurs in approximately 82% of HCC samples [132]. SOX17 inhibits the proliferation of HCC cells and inactivates the WNT/β-catenin signaling pathway through the HMG region [132].

The remarkably elevated expression of SOX18 has been observed in HCC tissues compared to that in adjacent nontumor tissues, particularly in cases of metastasis or recurrence [133,134]. Moreover, increased SOX18 expression is correlated with poor tumor differentiation, higher TNM stage, and worse prognosis of patients with HCC, which serves as an independent predictor for survival and recurrence [133,134]. Functionally, SOX18 promotes the proliferation, cell cycle process, EMT process, invasion, and migration and inhibits apoptosis of HCC [133,135]. SOX18 promotes cell viability partly through regulating the AMPK/mTOR signaling pathway [135]. On the basis of these findings, our previous study further found that SOX18 promotes the metastasis of HCC by activating the transcription of FGFR4 and FLT4 directly [134]. The overexpression of SOX18 in HCC can be attributed to the mechanism by which FGFR4 and its ligand, FGF19, activate the p-FRS2/p-GSK3β/β-catenin pathway to transactivate the SOX18 promoter directly [134]. Therefore, a positive feedback loop of FGF19/SOX18/FGFR4 is formed and facilitates the HCC metastasis [134].

SOX30 is the only member of the SOX H subgroup. To this point, the tumor suppressor role of SOX30 has been detected in multiple tumors, including bladder cancer and lung cancer [136,137]. In HCC tissues, the expression of SOX30 is decreased compared to adjacent non-tumor tissues [138]. Consistently, SOX30 interferes with cell proliferation and induces cell apoptosis by transactivating its downstream p53 directly in HCC [138]. Furthermore, SOX30 is also a target of miR-645 to mediate the role of miR-645 in promoting HCC progression [138].

## 4. SOX Transcription Factors and Tumor Immune Microenvironment

The TIME is a heterogenous ecosystem consisting of various innate and adaptive immune cells [139]. Substantial evidence suggests that suppressive TIME attributed to the crosstalk between different cells through cytokines, chemokines, and immunosuppressive checkpoints is frequently present in tumors and prominently accounts for the tumor initiation, progression, and therapeutic resistance [139,140,141]. It is becoming increasingly evident that SOX factors play a role in remodeling TIME by impacting multiple immune cells, including effector CD8+ T cells, neutrophils, B cells, and myeloid-derived suppressor cells (MDSCs).

### 4.1. The Role of SOX in Tumor Immune Microenvironment

Type I interferon (IFNI) signaling, in addition to fighting viral infection, also promotes the recruitment and activation of effector T-cells, activates antigen-presenting cells (APCs), and facilitates the cross-priming of dendritic cells (DCs) in the TME, therefore playing an essential role in natural and therapy-induced cancer immunosurveillance [142,143]. In head and neck squamous cell carcinoma, experimental data suggest that SOX2 induces immunosuppressive TME through disturbing the stimulator of interferon genes (STING)-mediated IFNI signaling activation [144]. In detail, SOX2 negatively regulates the transcription of IFNB1 and IFNI target CXCL10 induced by STING agonist cGAMP and intracellular poly (dA:dT), and accelerates autophagy-mediated STING degradation, leading to a decrease of CD8+T cells infiltration and a relative increase of PD-1^high^ CD8+T cells population, which represents an immune exhaustion state, eventually contributing to tumor growth [144]. Of note, the immune landscape of tumor specimens showed that the SOX2 expression is positively associated with the regulatory T cell (Treg) infiltration and is negatively related to M1-like macrophages, which is consistent with the fact that IFNI promotes M1-like polarization of APCs [144]. Consequently, this SOX2-IFNI axis is a pivotal signal to mediate tumor immune escape. Furthermore, the SOX2/IFN-mediated tumor immune escape has also been demonstrated in melanoma. Ruiyan Wu et al. reported that SOX2 inhibits the transcription of SOCS3 and PTPN1, further activates the JAK-STAT signaling pathway, and up-regulates the interferon-stimulated genes resistance signature (ISG.RS) to hinder the infiltration and cytotoxicity of CD8+T cells [145]. Evidence showed that IFNI signaling also induces resistance to immune checkpoint inhibitors (ICIs) by activating ISG.RS [146,147]. In their study, they found that high SOX2 expression is associated with worse survival and low objective response rate in patients with high PD-L1 who were treated with the anti-PD-1 monoclonal antibody, which indicates that SOX2 is an independent predictor for poor prognosis and resistance to anti-PD-1 treatment in melanoma patients with high PD-L1 levels [145]. By developing genetically engineered mouse models of non-small cell lung cancer, Gurkan Mollaoglu et al. found that the overexpressed SOX2 and the inhibition of NKX2-1 by SOX2 synergistically promote tumor-associated neutrophil (TANs) recruitment through up-regulating CXCL5 expression, facilitating adeno-to-squamous transdifferentiation and squamous tumorigenesis [148]. Additionally, in glioblastoma stem-like cells, SOX2 and OCT4 cooperate to promote immunosuppressive TME by activating immunosuppressive transcriptome mediated by the BRD4/H3k27Ac axis, which includes various immunosuppressive checkpoint molecules (i.e., PD-L1, CD70, A2aR, TDO), cytokines, and chemokines involved in T-cell apoptosis, Treg infiltration, and M2 macrophage polarization [149]. This finding provides a clue to the broader role of SOX2 in constructing immunosuppressive TME. However, conflicting results indicate that SOX2 inhibits the expression of PD-L1 during stem cell differentiation and lung cancer cell plasticity [150]. Consequently, the specific immunosuppressive roles and mechanisms of SOX2 in different tumor types remain to be elucidated.

In a triple-negative breast cancer (TNBC) model, SOX4 enables the induction of the resistance of cancer cells to cytotoxic T cells through regulating several innate and adaptive immune pathways, including the suppression of IFNI-stimulated genes and MHC class I pathway genes (HLA-A, HLA-B, and TAP1) [151]. It has been reported that TGFβ induces SOX4 expression [152,153]. In this research, a mechanism study found that ITGAV, which encodes integrin αv protein, up-regulates SOX4 expression by activating TGFβ from a latent precursor [151]. Therefore, the αvβ6–TGFβ–SOX4 pathway is essential in conferring cancer cell resistance to T cell-mediated cytotoxicity and serves as a promising therapeutic target for cancers. In addition, a previous study also revealed that SOX4 is a negative target of miR-132 in B cells, promoting B cell development and increasing the potential of B cell cancer occurrence [154].

Through comparatively analyzing the impact of distinct genetic backgrounds in prostate cancer on the composition of immune cells within TME, Marco Bezzi et al. uncovered that the loss of Zbtb7a up-regulates the CXCL5 expression by inducing SOX9 transcriptional activity, thus increasing the recruitment of polymorphonuclear MDSCs in the *Pten*-deficient tumors and facilitating tumor progression [155]. Likewise, Col1 deletion in aSMA^+^ myofibroblasts promotes the recruitment of CD206^+^F4/80^+^Arg1^+^ MDSCs through the SOX9-CXCL5 axis and further inhibits T and B lymphocyte function, thereby accelerating pancreatic ductal adenocarcinoma progression [156]. Consequently, the SOX9-CXCL5 axis represents a key regulatory mechanism that mediates the MDSCs chemotaxis to tumors and provides a possible therapeutic strategy for different tumors.

### 4.2. Association between the Expression of SOX and That of Immune Components in Hepatocellular Carcinoma

So far, the regulatory role of SOX factors in the TIME of HCC is still completely unclear. Herein, we initially explored the association between SOX expression and immune components in HCC using the TIMER2 database. As shown in Figure 3, the expression of most SOX members was significantly associated with the infiltration of eight immune cells in HCC, including CD8+ T cell, CD4+ T cell, B cell, macrophage (M1 and M2 types), MDSC, Treg, neutrophil, and DC. Among them, the tumor-associated macrophage (TAM), MDSC, Treg, regulatory dendritic cell (DCreg), neutrophil, and regulatory B cell (Breg) are essential immunosuppressive cells that determinate the initiation and progression of tumors and as potential therapeutic targets for tumors [157,158,159]. Our analysis results showed that SOX4, SOX11, SOX13, and SOX15 had the most significant and comprehensive positive correlations with these immunosuppressive cells in HCC. 

Next, we performed gene co-expression analysis between SOX members and immune-related genes in HCC. Via linking the mRNA levels of SOX factors with that of 47 well-known immune checkpoint genes in HCC [160], we found that most SOX members except SRY, SOX1, SOX3, SOX10, and SOX14 were significantly positively correlated with these immune checkpoints (Figure 4A). Of note, some members have shown remarkable correlation with classic immune checkpoints that have successfully implemented clinical translations, including PD-1 (encoded by *PDCD1*), PD-L1 (encoded by *CD274*), and CTLA4 (encoded by *CTLA4*). In addition, we also revealed that the expression of many SOX members was observably associated with that of human leukocyte antigen (HLA)-I and II molecular [161] (Figure 4B), chemokines (Figure 5A), and chemokines receptors (Figure 5B) in HCC.

Taken together, the expression correlation between SOX factors and immune components was close in HCC, suggesting that SOX factors were likely to participate in TIME remodeling of HCC and might act as potential biomarkers for predicting immunotherapy response in HCC. 

## 5. SOX Transcription Factors and Translational Potential

Although the crucial role of SOX factors in regulating the initiation and progression of tumors has been well documented, directly targeting these SOX proteins through inhibitors remains a challenge. The explanation for this difficulty is that, unlike the tractable catalytic sites of kinases, TF functions via protein-DNA or protein-protein interactions to form convex and highly positively charged DNA binding interfaces or to flatter protein binding surfaces, losing the structure of the deep pocket in kinases [4]. This structure makes the TFs ‘undruggable’ for traditional small-molecule inhibitors based on structure design. In this section, we summarize several alternative strategies for targeting SOX with translational potential in cancer therapy and detail the experimental evidence in multiple tumors (Table 1).

### 5.1. SOX Factors as Biomarkers for Patient Stratification Treatment

An alternative approach is to use SOX as a biomarker for patient stratification treatment; in other words, to target the upstream or downstream signal molecules of SOX for patients with high SOX levels.

In melanoma, the SOX2-BRD4 transcriptional complex has been verified to activate GLI1 via a noncanonical hedgehog/GLI signaling [162]. Smoothened (SMO) is a receptor that mediates downstream GLI1 activation in canonical hedgehog/GLI signaling. However, targeting SMO is frequently ineffective in multiple tumors due to the activation of noncanonical hedgehog/GLI signaling. Recently, researchers combined the SMO inhibitor MRT-92 and a Proteolysis Targeted Chimeras (PROTACs)-derived BRD4 degrader (MZ1), to treat the tumor and found that it remarkably suppressed tumor growth [162]. Simultaneously, high levels of SOX2 serve as a biomarker to achieve precise subpopulation selection for this promising therapeutic strategy [162].

In previous research, we have introduced that the integrin αvβ6–TGFβ–SOX4 pathway promotes immune evasion and tumor progression in TNBC [151]. They further found that integrin αvβ6-blocking antibody monotherapy or combined with a PD-1 antibody, which has entered clinical trials, significantly down-regulates the SOX4 expression, sensitize cancer cells to killing by CD8+ T cells, and reduce the primary tumor burden and lung metastatic burden [151]. This finding not only provides a promising immunotherapy strategy but also implies that SOX4 is a biomarker to guide this immunotherapy.

SOX9 also acts as an indicator for subpopulation treatment. Evidence showed that the WNT/β-catenin signaling is activated in prostate cancer cells, especially those with SOX9 overexpression, indicating that prostate cancer patients with high SOX9 levels are more sensitive to WNT pathway antagonists [163]. Furthermore, given the molecular mechanism by which SOX9 activates the PI3K/AKT/mTOR signaling pathway in esophageal cancer, the highly expressed SOX9 is a potential indicator for a subset of esophageal cancer patients who are more responsive to rapamycin, a specific mTOR inhibitor that has been used clinically [164].

In our previous study, on the basis of the finding that the FGF19-SOX18-FGFR4 positive feedback loop promotes HCC metastasis, we further investigated the effect of BLU9931, a specific FGFR4 inhibitor, in HCC [134]. Results showed that BLU9931 treatment remarkably abrogates the SOX18-induced invasion and metastasis of HCC, suggesting that SOX18 is a biomarker for identifying HCC patients who benefit from FGFR4 inhibitor treatment [134].

### 5.2. Targeting SOX Proteins Degradation

Though screening the epigenetic compounds library, researchers identified that suberoylanilide hydroxamic acid (SAHA), a histone deacetylase inhibitor that has been approved for the treatment of cutaneous T-cell lymphoma [175], facilitates SOX2 acetylation and proteasome-dependent degradation, therefore abolishing the SOX2-mediated resistance of tumor cells to CD8+T cells [145]. It is worth noting that the administration of SAHA significantly improves the therapeutic efficacy of the anti-PD-1 antibody in a mouse model of melanoma [145]. Currently, several clinical trials evaluating the efficacy of combining SAHA and ICIs are ongoing (NCT02638090, NCT02538510, NCT02619253, NCT02395627). In addition, ChlA-F, a conformation-derivative of Chel A isolated from *Goniothalamus cheliensis Hu*, promotes SOX2 ubiquitination and protein degradation by enhancing the mRNA stability of E3 ligase USP8 and also inhibits SOX2 protein translation by activating c-Jun-miR-200c, thus suppressing the invasion of bladder cancer cells [165]. However, the efficacy and safety of ChlA-F remain to be investigated in vivo.

### 5.3. SOX Factors as Peptide Vaccine Boost Anti-Tumor Immune Response

Choosing appropriate targeted tumor antigens to enhance the anti-tumor immune response mediated by T cells is an effective cancer treatment strategy. Accumulated evidence indicates that several SOX members have been identified as tumor-specific antigens that can augment cytotoxic T lymphocytes (CTLs) response to kill cancer cells, including SOX2 (glioblastoma) [166], SOX4 (lung cancer) [167], SOX6 (glioblastoma) [168,169], and SOX11 (glioblastoma) [170]. In general, these SOX factors are elected due to their significant overexpression in most tumor tissues and their almost negligible levels in nontumorous tissues.

### 5.4. Tumor-Targeted Delivery of siRNA to Silence SOX Expression

Using small interfering RNAs (siRNAs) for gene therapy has the advantages of easy synthesis, high binding specificity, and significant silencing efficacy [176]. However, the obstacles that include poor stability of RNA oligomers, off-target effects, and inability to cross biological barriers have greatly restricted their applications [176]. Delivery of siRNAs through vehicles is a promising strategy to solve the above impediments. Nanoparticles (NPs) as a carrier further provide advantages over other delivery vehicles [176]. Terrick Andey et al. designed a lipoplex nanoparticle to deliver therapeutic siRNA targeting SOX2 (CL-siSOX2) to a mouse xenograft lung cancer model [171]. They found that it significantly suppresses the growth of the tumor with high SOX2 levels and inhibits the expression of markers involved in tumor growth, metastasis, and chemoresistance [171]. Furthermore, in addition to being well-tolerant to CL-siSOX2, the mice also show fewer side effects and decreased tumor size in the combination treatment group of CL-siSOX2 and cisplatin, which provides an effective SOX2-targeted strategy for lung cancer, either monotherapy or in combination with cisplatin [171]. Additionally, another study improved the delivery system of selenium nanoparticles (SeNPs) and developed more stable RGDfC-modified functionalized SeNPs (RGDfC-SeNPs) to selectively deliver siSOX2 to HCC cells through clathrin-mediated endocytosis and releasing siSOX2 in the lysosome [172,177]. This RGDfC-SeNPs/siSOX2 complex exhibits a significant suppressive role on HCC with low toxicity in vitro and in vivo, indicating its potential for HCC-targeted therapy via silencing SOX2 [172].

### 5.5. Targeting Endogenous SOX Expression by Artificial Transcription Factors-Based Technologies

Artificial transcription factors (ATFs) are a molecular tool with great potential that can regulate endogenous target gene expression. Among them, zinc-finger (ZF)-based ATFs enable the specific and effective modulating of target genes expression in vitro and in vivo, which provides additional advantages [178]. In a previous study, researchers engineered three ATFs that bind to the proximal SOX2 promoter and one ATF that targets SOX2 regulatory region I (SRR1), one of which inhibits the expression of SOX2 in breast cancer cells by as much as 94% [173]. Some of these ZF-based ATFs significantly inhibit the growth of breast cancer in vitro and in vivo for a long time [173]. Mechanistically, these ZF-based ATFs play an inhibitory role on SOX2 expression through recruiting the transcriptional repressor domain Kruppel-Associated box (SKD domain) to the SOX2 promoter and further recruiting the co-repressor KAP1 to facilitate chromatin condensation mediated by histones deacetylation and H3K4me3 as well as H3K9me3 demethylation [173,179,180]. Similarly, targeting SOX2 by ZF-based ATFs has also been applied in lung and esophageal squamous cell carcinoma and remarkably inhibits the growth of tumors in vitro and in vivo [174]. Taken together, ZF-based ATFs are an effective molecular-targeted therapy to potentially address the ‘undruggable’ nature of SOX.

## 6. Discussion and Outlook

Over the past decades, not only the role of the SOX family in cell fate decisions has been elucidated, but also its relevance to the initiation and progression of tumors has become gradually clear. Substantial evidence reveals that the SOX family regulates the malignant phenotypes of various tumors extensively, including proliferation, migration, angiogenesis, CSCs properties, EMT, and drug resistance [181]. In HCC, in addition to impacting these phenotypes that are common in other tumors, SOX factors are also involved in the regulation of HCC-specific hallmarks. For example, HBV interacts with SOX4 in HCC, although it is still controversial whether SOX4 regulates HBV replication positively or negatively [64,65]. From a molecular mechanism standpoint, SOX factors play their tumor-promoting or tumor suppressing role in HCC through modulating multiple key signaling pathways in a transcription-dependent or independent manner, including Wnt/β-catenin signaling, TGFβ signaling, Notch signaling, AMPK/mTOR signaling, and p53 signaling. As the primary regulator of body development, the Wnt/β-catenin signaling pathway is the most extensive mediator that affects SOX factors to regulate HCC, especially mediating the role of SOX in regulating the liver CSCs properties [95,128]. Additionally, the aberrant expression of SOX factors is frequently observed in HCC tissues, and some of them act as potential prognostic factors in tumors. According to current studies, promoter hypermethylation, signaling pathway regulation such as TGFβ signaling, AKT signaling, and FGF19/p-FRS2/p-GSK3β/β-catenin signaling, and post-transcriptional modulation by miRNAs, are the main factors responsible for abnormal SOX expression. In summary, this compelling evidence emphasizes the key role of SOX members in the initiation and progression of HCC, but whether they have unknown functions and molecular mechanisms in HCC still needs to be explored.

The TIME is characterized by heterogeneity and suppression, and it acts as a key and the hottest factor affecting the initiation and development of tumors. In recent years, the relationship between SOX members and TIME has been gradually revealed. These SOX factors serve as important regulators to mediate the interaction between cancer cells, mesenchymal cells, and immune cells such as effector CD8+ T cells, neutrophils, B cells, and MDSCs, thus remodeling TIME and impacting tumor development. However, the effect of SOX members on TIME in HCC remains unknown so far. In this review, we initially analyzed the association between SOX expression and immune components in HCC by using the TIMER2 database. The exciting results showed that, except for SRY, SOX1, SOX10, and SOX14, the vast majority of SOX members were closely correlated with the infiltration of CD8+ T cell, CD4+ T cell, B cell, macrophage, MDSC, Treg, neutrophil, and DC in HCC (Figure 3). Furthermore, we analyzed the gene co-expression relationship between SOX members and immune checkpoint genes, antigen-presenting molecules, chemokines, and chemokine receptors in HCC and suggested that the expression of most SOX members was implicated in those of immune-related molecules (Figure 4 and Figure 5). In general, our results provided a vital clue that the role of SOX members on HCC extended to affect its TIME and paved the way for exploring the role and mechanism of SOX members in regulating the TIME of HCC by interacting with different immune cells.

Many SOX members show significant clinical implications in HCC. For instance, the overexpressed SOX4 or SOX12 is positively associated with the poor survival of HCC, while low SOX6 or SOX7 expression predicts shorter disease-free survival and overall survival, which provide potential diagnostic and prognostic markers for HCC [56,57,58,59,79,87,127]. In addition to serving as the predictors of diagnosis and prognosis, many SOX members are also potential therapeutic targets for HCC. However, despite these wide acknowledgments of the important role of SOX factors on HCC and the encouragement from some successful cases of other TFs in preclinical development or clinical trials (such as STAT TFs), advances in SOX-targeted therapies still face a great challenge. Some alternative strategies with translational potential that include targeting upstream or downstream signal molecules of SOX for patients with high SOX levels, targeting SOX proteins degradation, developing a SOX-related peptide vaccine, delivering siRNA-SOX, and targeting endogenous SOX expression by ATF-based technologies have been proposed in recent years and have obtained promising results in initial preclinical experiments. However, these approaches still have many shortcomings preventing their advancement. Therefore, it is necessary to adopt or develop reasonable and practical approaches for these ‘undruggable’ molecules. For example, the existing methods for SOX proteins degradation lack specificity and stability. A new approach termed PROTACs is designed to degrade polyubiquitin-labeled target proteins through the ubiquitin-proteasome system, thereby avoiding some limitations of traditional small molecule therapy strategies [182]. Although this approach is currently only developed for some ‘druggable’ molecules, it also provides an opportunity for the ‘undruggable’ targets, and future research should make efforts in this regard [183]. Additionally, the tertiary structure of proteins is essential for the development of targeted therapeutic drugs. A recent study found that the intrinsically disordered regions frequently observed in many TFs become structured after being folded and bound to their partners, thereby forming a ‘druggable’ structure [4]. This strategy encourages future research to explore the finer 3D structures of SOX proteins and the interaction between SOX factors and their binding partners.

## 7. Conclusions

In summary, the significance of SOX TFs in the initiation and development of HCC has been concerned and increasingly uncovered. Given that TIME is critical for tumor initiation and progression, we not only provided a first overview understanding of the remodeling effect of SOX factors on TIME in multiple tumors by summarizing the roles and regulatory mechanisms of SOX factors on immune cells in several tumors in detail, but also revealed the close relationship between SOX factors with eight immune cells and numerous immune-related molecules in HCC by bioinformatic analysis for the first time, which provided an initiatory and latest clue for exploring the effect of SOX factors on TIME in HCC and developing novel combination immunotherapy strategies. As intractable TFs, we concluded a series of alternative strategies being developed in preclinical or clinical trials for targeting SOX in tumors, and proposed some promising alternative strategies, providing some new ideas for SOX-targeted therapy. However, there is still a considerable challenge in turning the targeted SOX factors in cancer from undruggable to reality. Consequently, future work should continue to advance SOX-related research involving molecular mechanisms, anti-tumor immune response, and targeting therapies, such as combination immunotherapy.

## Figures and Tables

**Figure 1 cancers-14-01165-f001:**
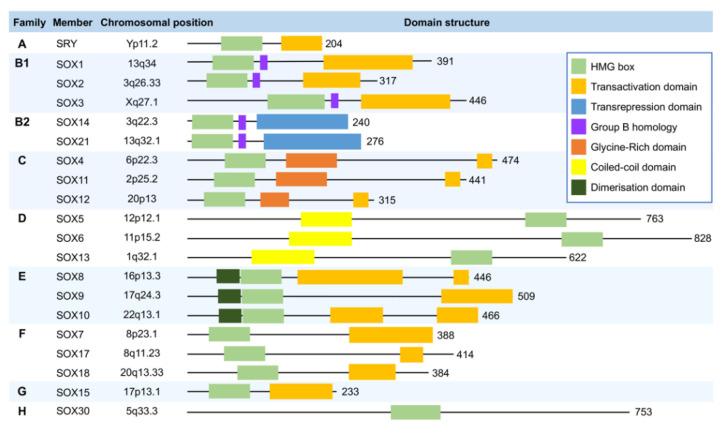
Schematic diagram of the chromosomal positions and domain structures of human SOX transcriptional factors. SOX family members are classified into eight subgroups based on the degree of sequence identity of the HMG motif, which is an evolutionarily conserved DNA-binding motif and is also a representative signature of the SOX family. Besides the shared HMG motif, different SOX members have their specific structure domains to achieve functional diversity. The chromosomal positions and domain structures of SOX members have been shown (Data referenced from Ref. [17] and website www.ncbi.nlm.nih.gov/gene/ (accessed on: 15 November 2021)). Abbreviations: SRY: Sex-determining region Y; SOX: Sex-determining region Y-related high-mobility group box; HMG: High-mobility group.

**Figure 2 cancers-14-01165-f002:**
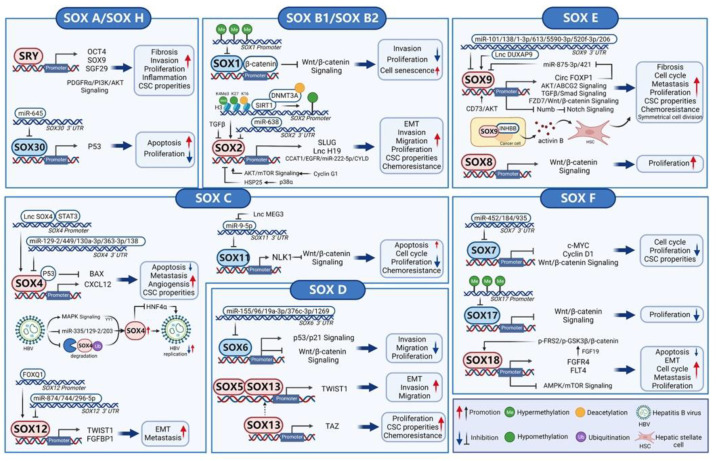
The roles and molecular mechanisms of SOX transcription factors in hepatocellular carcinoma (HCC). SOX members serve as master tumor drivers or suppressors to transcriptionally regulate key downstream targets or signaling pathways, therefore controlling the initiation and progression of HCC. Simultaneously, the dysregulated SOX expression is frequently observed in HCC, which is mainly attributed to promoter hypermethylation, signaling pathway regulation, and post-transcriptional modulation by miRNAs (microRNAs). This diagram was created by BioRender.com (accessed on: 28 November 2021). The red box represents tumor driver, and the blue box represents tumor suppressor. Abbreviations: SOX: Sex-determining region Y-related high-mobility group box; SRY: Sex-determining region Y; CSC: Cancer stem cell; 3′UTR: 3′ untranslated region; H3: Histone 3; EMT: Epithelial-mesenchymal transition.

**Figure 3 cancers-14-01165-f003:**
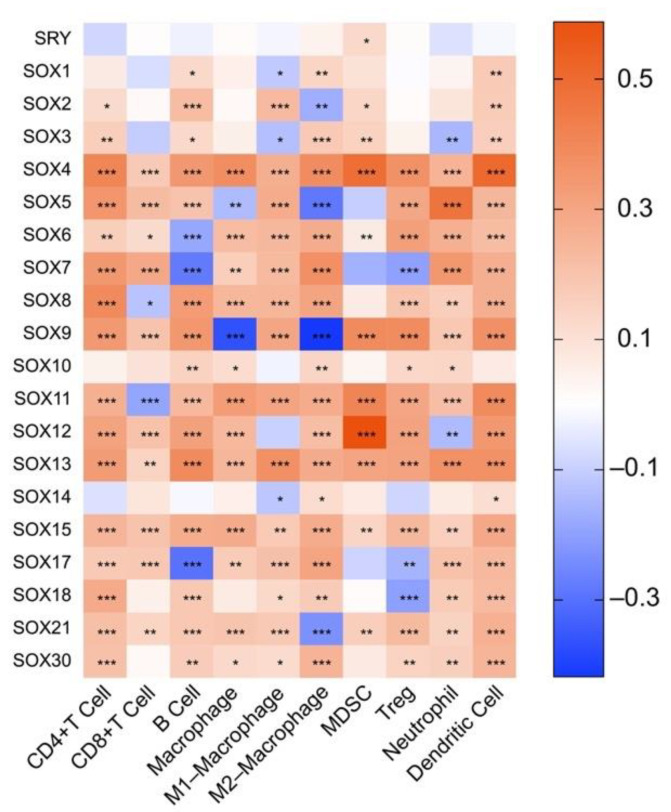
The association heatmaps between SOX transcription factor expression and immune cell infiltration in hepatocellular carcinoma. Red indicates positive correlation, and blue indicates negative correlation. The darker the color, the stronger the correlation. The data was analyzed by the TIMER2. * *p* < 0.05, ** *p* < 0.01, *** *p* < 0.001.

**Figure 4 cancers-14-01165-f004:**
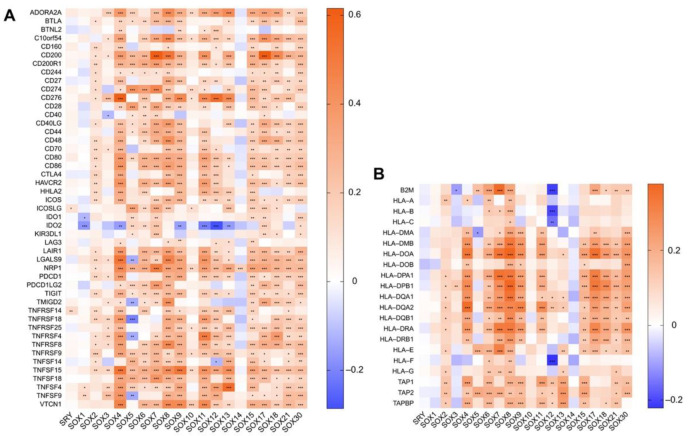
Correlation between the expression of SOX transcription factors and that of immune-related genes in hepatocellular carcinoma. (**A**) The relationships between SOX factors expression and the levels of immune checkpoints genes and (**B**) antigen-presenting molecules were analyzed by the TIMER2. Red indicates positive correlation and blue indicates negative correlation. The darker the color, the stronger the correlation. * *p* < 0.05, ** *p* < 0.01, *** *p* < 0.001.

**Figure 5 cancers-14-01165-f005:**
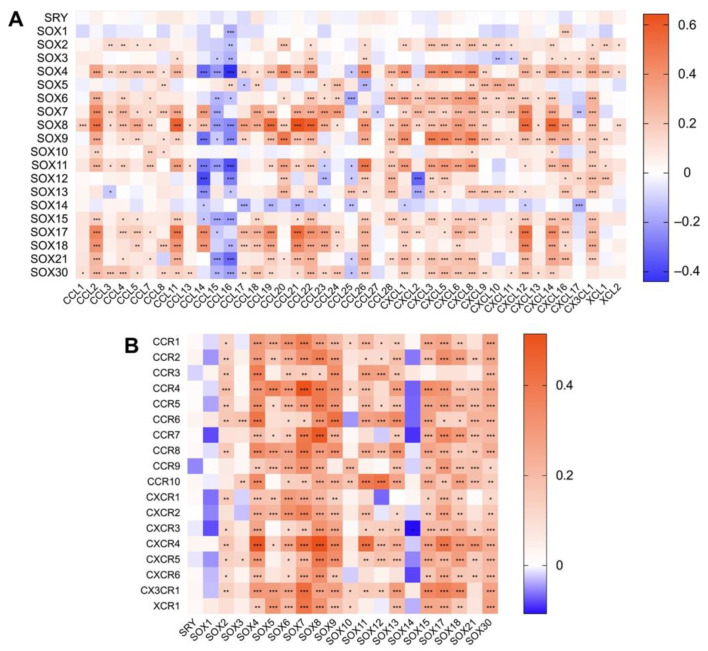
Correlation between the expression of SOX transcription factors and that of immune cell chemotaxis-related genes in hepatocellular carcinoma. (**A**) The relationships between SOX factors expression and the levels of chemokines and (**B**) chemokine receptors were analyzed by the TIMER2. Red indicates positive correlation and blue indicates negative correlation. The darker the color, the stronger the correlation. * *p* < 0.05, ** *p* < 0.01, *** *p* < 0.001.

**Table 1 cancers-14-01165-t001:** The alternative strategies for targeting SOX factors in cancers.

Strategy	Target	Biomarker	Cancer	Mechanism of Action	Clinical Trial
SOX factors as biomarkers for patient stratification treatment					
MRT-92 plus MZ1 [162]	SMO+ BRD4	SOX2	Melanoma	SMO and SOX2-BRD4 complex activate hedgehog/GLI signaling in canonical and noncanonical manners, respectively	/
Integrin αvβ6/8 mAb [151]	Integrin αvβ6/8	SOX4	TNBC	Integrin αv up-regulates SOX4 by activating TGFβ from a latent precursor	Phase II NCT03688230
LGK974 [163]	WNT	SOX9	Prostate cancer	SOX9 reactivates the WNT/β-catenin signaling	Phase I NCT01351103
Rapamycin [164]	mTOR	SOX9	Esophageal cancer	SOX9 inhibits miR-203a to activate the PI3K/AKT/mTOR signaling	Phase III NCT04736589
BLU9931 [134]	FGFR4	SOX18	HCC	SOX18 transactivates FGFR4 and FLT4, and FGFR4-FGF19 in turn up-regulates SOX18	/
Targeting SOX proteins degradation					
SAHA [145]	SOX2	SOX2	Melanoma	Promoting SOX2 acetylation and proteasome-dependent degradation	Phase I/II NCT02638090
ChlA-F [165]	SOX2	SOX2	Bladder cancer	promoting SOX2 ubiquitination and protein degradation by enhancing the mRNA stability of USP8 and inhibiting SOX2 protein translation by activating c-Jun-miR-200c	/
SOX factors as peptide vaccine boost anti-tumor immune response					
SOX2-derived peptide [166]	SOX2	SOX2	Glioblastoma	As a tumor-specific vaccine antigen to augment CTLs response	Phase I NCT02157051
SOX4-derived peptide [167]	SOX4	SOX4	Lung cancer	As a tumor-specific vaccine antigen to augment CTLs response	/
SOX6-derived peptide [168,169]	SOX6	SOX6	Glioblastoma	As a tumor-specific vaccine antigen to augment CTLs response	/
SOX11-derived peptide [170]	SOX11	SOX11	Glioblastoma	As a tumor-specific vaccine antigen to augment CTLs response	/
Tumor-targeted delivery of siRNA to silence SOX expression					
CL-siSOX2 [171]	SOX2	SOX2	Lung cancer	Deliver therapeutic siRNA targeting SOX2 to tumor in vivo through a lipoplex nanoparticle	/
RGDfC-SeNPs-siSOX2 [172]	SOX2	SOX2	HCC	Deliver therapeutic siRNA targeting SOX2 to tumor in vivo through a RGDfC-SeNP system	/
Targeting endogenous SOX expression by artificial transcription factors-based technologies					
ZF-552SKD [173]	SOX2 promoter	SOX2	Breast cancer	ZF-based ATF inhibits SOX2 expression through recruiting transcriptional repressor SKD domain to the SOX2 promoter and recruiting co-repressor KAP1 to facilitate chromatin condensation	/
ZF-598SKD [173]	SOX2 promoter	SOX2	Breast cancer	ZF-based ATF inhibits SOX2 expression through recruiting transcriptional repressor SKD domain to the SOX2 promoter and recruiting co-repressor KAP1 to facilitate chromatin condensation	/
ZF-619SKD [173]	SOX2 promoter	SOX2	Breast cancer	ZF-based ATF inhibits SOX2 expression through recruiting transcriptional repressor SKD domain to the SOX2 promoter and recruiting co-repressor KAP1 to facilitate chromatin condensation	/
ZF-4203SKD [173]	SOX2 regulatory region I	SOX2	Breast cancer	ZF-based ATF inhibits SOX2 expression through recruiting transcriptional repressor SKD domain to the SOX2 promoter and recruiting co-repressor KAP1 to facilitate chromatin condensation	/
ATF/SOX2 [174]	SOX2 promoter	SOX2	Lung and esophageal SCC	ZF-based ATF inhibits SOX2 expression through targeting the SOX2 distal and proximal promoter region and a KOX transcriptional repressor domain	/

Abbreviations: SMO: Smoothened; SOX: Sex-determining region Y-related high-mobility group box; mAb: Monoclonal antibody; TNBC: Triple-negative breast cancer; HCC: Hepatocellular carcinoma; SAHA: Suberoylanilide hydroxamic acid; CTLs: Cytotoxic T lymphocytes; CL-siSOX2: siSOX2 delivered by cationic lipoplex; RGDfC-SeNP: RGDfC-modified functionalized selenium nanoparticles; ZF-based ATF: Zinc-finger-based Artificial transcription factors; SKD: Kruppel Associated box; SCC: Squamous cell carcinoma;.

## Data Availability

Publicly available datasets were analyzed in this study. This data can be found here: [timer.cistrome.org/ (accessed on: 25 November 2021) and clinicaltrials.gov/ (accessed on: 28 November 2021)].

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
