# Peer review of "Advance of SOX Transcription Factors in Hepatocellular Carcinoma: From Role, Tumor Immune Relevance to Targeted Therapy"

_cancers, 2022, doi:10.3390/cancers14051165_

Round 1

Reviewer 1 Report

The research question is clearly outlined. The variables are well defined and measured appropriately. The study methods are valid and reliable. There are enough details provided in order to replicate the study. The data is presented in an appropriate way. The conclusions are supported by references and own results. The authors also discuss the future perspectives on using these knowledge for therapy development.

Specific comments on weaknesses of the article and what could be improved:

Major points  - none

Minor points

  1. Please, state the limitations of the study
  2. Could you please discuss the clinical implications of the results, except the therapy; for example, for diagnostic, follow-up, etc.

Author Response

Point 1: Please, state the limitations of the study

Response 1: The limitations of the study are as follows: 1. Numerous master molecules or signaling pathways are essential for maintaining body homeostasis and normal physiological processes. The abnormal or dysregulation of them often leads to disorders. Therefore, identifying the key role of molecules in physiological functions is a vital step and prerequisite for exploring their pathogenic potential and enabling precision therapy. Our study mainly focused on the roles of each SOX member in hepatocellular carcinoma (HCC) development, but lacked a detailed summary of their main physiological roles, and thus may hinder a profound and comprehensive understanding of how SOX members affect the disease progression, especially in HCC, and even influence the development of targeted therapy strategies, such as side effects. 2. Given the significance of the tumor immune microenvironment in tumor development, we preliminarily analyzed the correlation between the SOX members expression and immune cells infiltrations or key immune-related molecules expression in HCC using the TIMER2 database. However, there is a lack of in vitro and in vivo experiments to confirm the positive results of our work to further explore the regulatory mechanisms and provide more potential immunotherapy strategies.

Point 2: Could you please discuss the clinical implications of the results, except the therapy; for example, for diagnostic, follow-up, etc.

Response 2: Through reviewing extensive literature related to the SOX family, we summarized the roles and clinical implications of SOX members in HCC. For their clinical implications, in addition to being the potential therapeutic targets, many SOX members may also be potential diagnostic and prognostic markers for HCC. Specifically, the SOX4 expression is remarkably increased in HCC tissues, especially in metastatic and recurrent samples [1-7]. The overexpressed SOX4 is positively associated with higher intratumoral microvessel density, distant metastasis, and poor survival of HCC cases, which provides a potential diagnostic and prognostic marker for HCC [4-7]. In addition, the low SOX6 or SOX7 expression predicts shorter disease-free survival and overall survival, indicating that SOX6 and SOX7 are potential prognostic markers for HCC [8, 9]. SOX12 has been demonstrated to be a key pro-metastatic molecule in HCC in our previous study [10]. Meanwhile, we also revealed that SOX12 is significantly up-regulated in patients with metastasis or recurrence of HCC than patients without metastasis or recurrence [10]. The overexpression of SOX12 is associated with loss of tumor encapsulation, microvascular invasion, a higher TNM stage, shorter overall survival times, and more frequent recurrence rates, suggesting that SOX12 is an independent risk factor for poor survival and recurrence of HCC patients [10]. In conclusion, many SOX members have significant clinical values, whether as potential therapeutic targets or as diagnostic and prognostic markers for HCC, with potential for clinical translation in the future. The modifications have been made and highlighted in yellow in the revised version (page 19, line 766).

  1. Liao YL, Sun YM, Chau GY, Chau YP, Lai TC, Wang JL et al. Identification of SOX4 target genes using phylogenetic footprinting-based prediction from expression microarrays suggests that overexpression of SOX4 potentiates metastasis in hepatocellular carcinoma. Oncogene. 2008; 27: 5578-5589.
  2. Hur W, Rhim H, Jung CK, Kim JD, Bae SH, Jang JW et al. SOX4 overexpression regulates the p53-mediated apoptosis in hepatocellular carcinoma: clinical implication and functional analysis in vitro. Carcinogenesis. 2010; 31: 1298-1307.
  3. Chen X, Zhang L, Zhang T, Hao M, Zhang X, Zhang J et al. Methylation-mediated repression of microRNA 129-2 enhances oncogenic SOX4 expression in HCC. Liver Int. 2013; 33: 476-486.
  4. Chen ZZ, Huang L, Wu YH, Zhai WJ, Zhu PP, Gao YF. LncSox4 promotes the self-renewal of liver tumour-initiating cells through Stat3-mediated Sox4 expression. Nat Commun. 2016; 7: 12598.
  5. Sandbothe M, Buurman R, Reich N, Greiwe L, Vajen B, Gurlevik E et al. The microRNA-449 family inhibits TGF-beta-mediated liver cancer cell migration by targeting SOX4. J Hepatol. 2017; 66: 1012-1021.
  6. Tsai CN, Yu SC, Lee CW, Pang JS, Wu CH, Lin SE et al. SOX4 activates CXCL12 in hepatocellular carcinoma cells to modulate endothelial cell migration and angiogenesis in vivo. Oncogene. 2020; 39: 4695-4710.
  7. Huang JL, Wang XK, Liao XW, Han CY, Yu TD, Huang KT et al. SOX4 as biomarker in hepatitis B virus-associated hepatocellular carcinoma. J Cancer. 2021; 12: 3486-3500.
  8. Guo X, Yang M, Gu H, Zhao J, Zou L. Decreased expression of SOX6 confers a poor prognosis in hepatocellular carcinoma. Cancer Epidemiol. 2013; 37: 732-736.
  9. Wang J, Zhang S, Wu J, Lu Z, Yang J, Wu H et al. Clinical signi fi cance and prognostic value of SOX7 expression in liver and pancreatic carcinoma. Mol Med Rep. 2017; 16: 499-506.
  10. Huang W, Chen Z, Shang X, Tian D, Wang D, Wu K et al. Sox12, a direct target of FoxQ1, promotes hepatocellular carcinoma metastasis through up-regulating Twist1 and FGFBP1. Hepatology. 2015; 61: 1920-1933.

Reviewer 2 Report

Xiangyuan Luo uncovered SOX transcription factors in hepatocellular carcinoma (HCC) to be potentially relevant in tumour immune target therapy.

Point to be considered

1) The rationale of why the authors came up with this innovation.

2) What is the information that is not exactly available that motivated the authors to come up with this information. What are the current caveats and how do the authors highlight the current research in answering them? If not they need to address in future directions.

3) The authors need to highlight what new information the review is providing to enhance the research in progress.

4) Describe how the author's revision goes beyond the state-of-the-art, and the extent to the proposed work is ambitious for the next step in the research field.

5) The underlying message is that more precision and individualized approaches need to be tested. A challenge, but I would be interested in their perspective of how this might be done (i.e. the authors state that "Cyclin G1 up-regulates the expression of SOX2 by activating the Akt/mTOR signaling pathway, which leads to an increased proportion and enhanced tumour-initiating capacity of liver TICs, as well as induces chemoresistance of HCC cells to sorafenib and reference [39]; this reviewer personally misses some insights regarding a novel aspect of sorafenib sensitivity resistance).

For several years, sorafenib, a tyrosine kinase inhibitors (TKI) inhibitor, has been the approved treatment option, to date, for advanced HCC patients. Its activity is the inhibition of the retrovirus-associated DNA sequences protein (RAS)/Rapidly Accelerated Fibrosarcoma protein (RAF)/mitogen-activated and extracellular-signal regulated kinase (MEK)/extracellular-signal regulated kinases (ERK) signaling pathway. However, the efficacy of sorafenib is limited by the development of drug resistance, and the major neuronal isoform of RAF, BRAF and MEK pathways play a critical and central role in HCC escape from TKIs activity. Advanced HCC patients with a BRAF mutation display a multifocal and/or more aggressive behavior with resistance to TKI (please refer to PMID: 31766556 and expand).

Author Response

Point 1: The rationale of why the authors came up with this innovation.

Response 1: Hepatocellular carcinoma (HCC) is one of the deadliest health burdens worldwide and leads to dismal outcomes because of insufficient early diagnosis and few available treatment options for patients with advanced-stage HCC. Therefore, it is imperative to explore the molecular mechanisms and develop therapeutic strategies for HCC. Transcription factors (TFs) are pivotal sequence-specific DNA-binding proteins responsible for the first step of decoding DNA sequences, thus guiding the expression of the genome. Many TFs, which account for about 8% of all human genes, serve as master regulators to maintain normal physiological processes, such as the development process, and their dysregulation is closely associated with the disease. Over 2000 TFs divided into various families based on the homologous DNA-binding domain have been identified. Among them, SOX TF family, as cell fate determinants, plays a crucial role in diseases, especially in cancer. Although accumulating studies have explored the effect of SOX members on HCC, however, this evidence is fragmentary and partial. Our work is the first to summarize and integrate current research on the SOX family in HCC, including their roles and targeted therapeutic strategies, which appeals to the current needs in this direction and aims to point out the challenges we face and potential solutions. In addition, substantial evidence suggests that suppressive tumor immune microenvironment (TIME) attributed to the crosstalk between different cells is frequently present in tumors and prominently affects tumor initiation, progression, and therapeutic resistance. Besides, immunotherapy has shown significant advantages in anti-tumor therapy in recent years. Consequently, we also summarized the current study progress of SOX members in TIME. As HCC is characterized by immunologically privileged organ and immunogenic tumor, exploring its TIME and related molecular mechanisms is extremely important. To date, the regulatory role of SOX members in the TIME of HCC is still completely unclear. Therefore, we preliminarily analyzed the association between SOX expression and immune components in HCC using the TIMER2 database, aiming to provide a clue that the role of SOX members on HCC extended to affect its TIME and to pave the way for exploring the role and mechanism of SOX members in regulating TIME of HCC, and thus ultimately enable new immunotherapy strategies.

Point 2: What is the information that is not exactly available that motivated the authors to come up with this information. What are the current caveats and how do the authors highlight the current research in answering them? If not they need to address in future directions.

Response 2: The high mortality and few treatment strategies available of HCC make it an urgent task to study the mechanism of its initiation and development and new therapeutic targets. There have been quite a few studies that revealed the key tumor-driving or suppressing functions and related molecular mechanisms of SOX family in HCC according to our search, suggesting its crucial effect and possible translational potential on HCC. However, there is currently no review collecting and integrating studies of SOX in HCC. In addition, TIME plays a crucial role in tumor initiation and development. Immunotherapy is also one of the most successful strategies in current anti-tumor therapy, including in HCC. Nevertheless, the study of SOX in the TIME of HCC remains blank. Therefore, we focused on the roles and mechanisms of SOX factors in HCC and in the TIME of HCC. Through a search of existing studies, we found that most of the studies focused on the intracellular regulatory mechanism of SOX factors on HCC, while the role of SOX factors between HCC cells and the tumor microenvironment, especially TIME, was less studied. Additionally, the structure of SOX factors makes them “undruggable” for traditional small-molecule inhibitors based on structure design. Current research has proposed several alternative strategies for targeting SOX with translational potential in cancer therapy. However, there is still a considerable challenge in turning the targeted SOX factors in cancer from undruggable to reality. Therefore, these caveats should be concerned, and future research directions should concentrate on the development of the role of SOX factors in TIME and the novel targeted therapeutic strategy, such as combination immunotherapy.

Point 3: The authors need to highlight what new information the review is providing to enhance the research in progress.

Response 3: Thanks for the constructive comment. As suggested by the reviewer, we have highlighted the new information provided by our manuscript in the revised version (page 19, line 794). “In summary, the significance of SOX TFs in the initiation and development of HCC has been concerned and increasingly uncovered. Given that TIME is critical for tumor initiation and progression, we not only provided a first overview understanding of the remodeling effect of SOX factors on TIME in multiple tumors by summarizing the roles and regulatory mechanisms of SOX factors on immune cells in several tumors in detail, but also revealed the close relationship between SOX factors with eight immune cells and numerous immune-related molecules in HCC by bioinformatic analysis for the first time, which provided an initiatory and latest clue for exploring the effect of SOX factors on TIME in HCC and developing novel combination immunotherapy strategies. As intractable TFs, we concluded a series of alternative strategies being developed in preclinical or clinical trials for targeting SOX in tumors, and proposed some promising alternative strategies such as PROTACs, providing some new ideas for SOX-targeted therapy.”

Point 4: Describe how the author's revision goes beyond the state-of-the-art, and the extent to the proposed work is ambitious for the next step in the research field.

Response 4: Although a small number of studies have summarized the role of SOX family in physiology or other diseases, our work differs in some ways and has unique advantages. First, HCC is one of the deadliest health burdens globally, and the crucial role and possible translational potential of SOX factors in HCC have been uncovered. However, there is currently no review related to the role of SOX in HCC. Our work is the first to summarize the roles and molecular mechanisms of SOX in HCC in detail and comprehensively. Second, the TIME is critical for tumor initiation, progression, and therapeutic resistance. Immunotherapy is also one of the most successful strategies in current anti-tumor therapy. Our review not only elaborated the roles and mechanisms of SOX factors in TIME of multiple solid tumors based on all relevant studies, but also analyzed the correlation of SOX factors expression with immune cells infiltrations and immune-related molecules expression in HCC for the first time by using the TIMER2 database. Finally, although SOX factors are 'undruggable' in tumors, we focused on SOX-targeted therapy in tumors and more comprehensively and systematically summarized several alternative therapeutic strategies. Consequently, our review not only provides an overview understanding of the roles, mechanisms, and targeted therapy of SOX family in HCC based on existing studies, but also provides new information on the role of the SOX family in TIME of HCC, which contributes to the future research on SOX factors in the TIME and the development of combination immunotherapy.

Point 5: The underlying message is that more precision and individualized approaches need to be tested. A challenge, but I would be interested in their perspective of how this might be done (i.e. the authors state that "Cyclin G1 up-regulates the expression of SOX2 by activating the Akt/mTOR signaling pathway, which leads to an increased proportion and enhanced tumour-initiating capacity of liver TICs, as well as induces chemoresistance of HCC cells to sorafenib and reference [39]; this reviewer personally misses some insights regarding a novel aspect of sorafenib sensitivity resistance).

For several years, sorafenib, a tyrosine kinase inhibitors (TKI) inhibitor, has been the approved treatment option, to date, for advanced HCC patients. Its activity is the inhibition of the retrovirus-associated DNA sequences protein (RAS)/Rapidly Accelerated Fibrosarcoma protein (RAF)/mitogen-activated and extracellular-signal regulated kinase (MEK)/extracellular-signal regulated kinases (ERK) signaling pathway. However, the efficacy of sorafenib is limited by the development of drug resistance, and the major neuronal isoform of RAF, BRAF and MEK pathways play a critical and central role in HCC escape from TKIs activity. Advanced HCC patients with a BRAF mutation display a multifocal and/or more aggressive behavior with resistance to TKI (please refer to PMID: 31766556 and expand).

Response 5: Thanks for the constructive comment. We studied and cited the review suggested by the reviewer (PMID: 31766556) and other literature, and discussed some of our insights regarding a novel aspect of sorafenib sensitivity resistance in the revised version(page 6, line 228). Specifically, “Of interest, blocking the Akt/mTOR signaling pathway by inhibitors significantly enhances the sensitivity of HCC cells to sorafenib, providing a potential combination therapeutic strategy for HCC [1]. Indeed, drug resistance has severely limited the efficacy of sorafenib. As a tyrosine kinase inhibitor (TKI) that mainly acts by inhibiting the RAS/RAF/MEK/ERK signaling pathway, the main neuronal isoform of RAF, BRAF, and MEK pathways and BRAF mutation are important resistance mechanisms of sorafenib [2]. Therefore, combination therapy strategies to eliminate these key resistance factors are the current priority. The activated Akt/mTOR signaling pathway has been demonstrated to confer sorafenib resistance to HCC through multiple mechanisms, including induction of Warburg shift [3] and increased TIC features [4]. The finding that SOX2-mediated the Akt/mTOR signaling pathway induces stemness and sorafenib resistance in HCC complements the stemness-related molecular mechanisms of sorafenib resistance, and further confirms the combination therapy of Akt/mTOR signaling pathway inhibitor and sorafenib for HCC is feasible.”

  1. Wen W, Han T, Chen C, Huang L, Sun W, Wang X et al. Cyclin G1 expands liver tumor-initiating cells by Sox2 induction via Akt/mTOR signaling. Mol Cancer Ther. 2013; 12: 1796-1804.
  2. Gnoni A, Licchetta A, Memeo R, Argentiero A, Solimando AG, Longo V et al. Role of BRAF in Hepatocellular Carcinoma: A Rationale for Future Targeted Cancer Therapies. Medicina (Kaunas). 2019; 55.
  3. Krstic J, Reinisch I, Schindlmaier K, Galhuber M, Riahi Z, Berger N et al. Fasting improves therapeutic response in hepatocellular carcinoma through p53-dependent metabolic synergism. Sci Adv. 2022; 8: eabh2635.
  4. Loh JJ, Li TW, Zhou L, Wong TL, Liu X, Ma VWS et al. FSTL1 Secreted by Activated Fibroblasts Promotes Hepatocellular Carcinoma Metastasis and Stemness. Cancer Res. 2021; 81: 5692-5705.

Reviewer 3 Report

Xiangyuan Luo et al uncovered the advance of SOX transcription factors in hepatocellular carcinoma, focusing on the role of tumour immune relevance to targeted therapy.

Missing points:

1 ‐ State of the art highlights, with a focus on novel insights regarding the tumor immunemicroenvironment. 

2 – A detailed description of the methodologies, objectives and results that the manuscript aimed to achieve and its result for the advancement of knowledge (along with a graphical abstract?)

3 ‐  identification of the role of each research unit, with regards to related modalities of integration and collaboration (Max. 10.000 characters)

4 – Possible application potentialities and scientific and/or technological and/or clinical impact (workflow figure?)

5 - The authors mentioned sorafenib and appropriately mentioned ref 39, 93, 94. This reviewer personally misses some insights regarding the fact that for several years, sorafenib, has been the approved treatment option, to date, for advanced HCC patients. Its activity is the inhibition of the retrovirus-associated DNA sequences protein (RAS)/Rapidly Accelerated Fibrosarcoma protein (RAF)/mitogen-activated and extracellular-signal regulated kinase (MEK)/extracellular-signal regulated kinases (ERK) signaling pathway. However, the efficacy of sorafenib is limited by the development of drug resistance, and the major neuronal isoform of RAF, BRAF and MEK pathways play a critical and central role in HCC escape from TKIs activity. Advanced HCC patients with a BRAF mutation display a multifocal and/or more aggressive behavior with resistance to TKI (please expand referring PMID: 31766556).

Author Response

Point 1: State of the art highlights, with a focus on novel insights regarding the tumor immune microenvironment.

Response 1: We have stated the highlights of our manuscript regarding the tumor immune microenvironment (TIME) in the revised version (page 19, line 795). Specifically, “Given that TIME is critical for tumor initiation and progression, we not only provided a first overview understanding of the remodeling effect of SOX factors on TIME in multiple tumors by summarizing the roles and regulatory mechanisms of SOX factors on immune cells in several tumors in detail, but also revealed the close relationship between SOX factors with eight immune cells and numerous immune-related molecules in HCC by bioinformatic analysis for the first time, which provided an initiatory and latest clue for exploring the effect of SOX factors on TIME in HCC and developing novel combination immunotherapy strategies.”

Point 2: A detailed description of the methodologies, objectives and results that the manuscript aimed to achieve and its result for the advancement of knowledge (along with a graphical abstract?)

Response 2: Methodologies: We performed a comprehensive literature search on the PubMed database for all studies on SOX transcription factor (TF) structure, SOX-related studies in hepatocellular carcinoma (HCC), SOX-related studies in the tumor immune microenvironment (TIME), and SOX-targeted therapy in HCC. We selected appropriate studies based on literature quality and publication year and summarized the roles and mechanisms of SOX factors in HCC, the regulatory mechanisms of SOX factors in TIME, and the SOX-targeted therapy strategies in tumors. To explore the immune correlation of SOX factors in HCC, we analyzed the association between SOX factors expression and immune cells infiltrations and immune-related molecules expression in HCC by the TIMER2 database.

Objectives: To provide a comprehensive understanding of the roles, molecular mechanisms, and targeted therapy of SOX factors in HCC. To provide an overview of the remodeling effect of SOX factors on TIME in multiple tumors and a preliminary relationship between SOX factors and immune signatures in HCC.

Results: The SOX family is divided into eight subgroups, some of which act as master tumor drivers or suppressors to transcriptionally regulate key downstream targets or signaling pathways, therefore controlling the proliferation, migration, angiogenesis, cancer stem cells (CSCs) properties, and drug resistance of HCC extensively. Aberrant expression of SOX factors is frequently observed in HCC tissues, which is mainly attributed to promoter hypermethylation, signaling pathway regulation, and post-transcriptional modulation by miRNAs. In TIME, these SOX factors serve as important regulators to mediate the interaction between cancer cells, mesenchymal cells, and immune cells such as effector CD8+ T cells, neutrophils, B cells, and MDSCs, thus remodeling TIME and impacting tumor development. In HCC, except for SRY, SOX1, SOX10, and SOX14, the vast majority of SOX factors were closely correlated with the infiltration of CD8+ T cell, CD4+ T cell, B cell, macrophage, MDSC, Treg, neutrophil, and DC. In addition, the expression of most SOX factors was associated with the immune-related molecules in HCC, including immune checkpoint genes, antigen-presenting molecules, chemokines, and chemokine receptors. As intractable TFs, a series of alternative strategies being developed in preclinical or clinical trials for targeting SOX in tumors have been described. Furthermore, some novel SOX-targeted strategies have been proposed.

In summary, our review integrated the fragmentary information involved in the effects of SOX factors on HCC and the SOX-targeted therapy strategies in tumors, providing a comprehensive and profound understanding of the roles and molecular mechanisms of SOX factors in HCC, and contributing to the advance in SOX-targeted therapy in tumors. Furthermore, we provided an overview of the remodeling effect of SOX factors on TIME in multiple tumors and a preliminary relationship between SOX factors and immune signatures in HCC, which paved the way for exploring the effect of SOX factors on TIME in HCC and developing novel combination immunotherapy strategies.

Point 3: identification of the role of each research unit, with regards to related modalities of integration and collaboration (Max. 10.000 characters)

Response 3: Our review summarizes and introduces the SOX family in tumors, especially in HCC, from the aspects of structure, role and molecular mechanism, as well as treatment. The structure determines the nature, and the nature determines the use. Therefore, in addition to the "Introduction", we first describe the structural features of the SOX family in the " Overview of the SOX Transcription Factors" section of the manuscript. This section introduces that an evolutionarily conserved DNA-binding motif termed HMG box is the representative signature of the SOX family, and the sequence RPMNAFMVW of the HMG box is the most extensive signature that can identify SOX genes. On the basis of the degree of sequence identity of the HMG motif, SOX members are classified into eight subgroups (SOX A/B/C/D/E/F/G/H). In general, SOX proteins in the same subgroup equip with similar biochemical properties and probably perform repetitive functions, while SOX proteins from different subgroups play distinct and even opposed functions. The functional diversity depends on their structural differences. Furthermore, identifying the physiological roles of molecules is a vital step and prerequisite for exploring their pathogenic potential and enabling precision therapy. Therefore, we also introduce the major physiological functions of SOX proteins and show that they play essential roles in cell fate decisions involving numerous developmental processes, especially in controlling stem and progenitor cell fate and tissues regeneration. Taken together, this section presents the structure, classification, and physiological function of SOX family, which will contribute to better understand how the SOX family functions in disease, especially in cancer.

Given that HCC is one of the deadliest human health burdens worldwide, in the next section of our manuscript, “SOX Transcription Factors in Hepatocellular Carcinoma”, we comprehensively describe the expression patterns, roles, and molecular mechanisms of SOX family in HCC. We introduce them according to different subgroups of the SOX family since SOX factors in the same subgroup generally show similar functional characteristics. However, some SOX factors belonging to the same subgroup also present opposite roles in the initiation and progression of HCC, and we compared the functional similarities and differences of these SOX factors in detail. Furthermore, this section also summarizes the clinical significance of SOX factors in HCC, including some of them as diagnostic and prognostic markers or therapeutic targets. In summary, this section is one of the most important parts of our manuscript, demonstrating the crucial functions, specific molecular mechanisms, and clinical significance of SOX factors in HCC, providing a research basis of the SOX-targeted therapy strategy for HCC.

Substantial evidence suggests that suppressive TIME attributed to the crosstalk between different cells is frequently present in tumors and prominently accounts for the tumor initiation, progression, and therapeutic resistance. Furthermore, immunotherapy is one of the most successful strategies in current anti-tumor therapy. Consequently, we specifically summarize the research progress of SOX factors in TIME in the “SOX Transcription Factors and tumor immune microenvironment” section of our manuscript. First, we dissect the role and molecular mechanism of SOX factors in TIME of all solid tumors based on existing studies. This evidence suggests that SOX factors play a role in remodeling TIME by impacting multiple immune cells, including effector CD8+T cells, neutrophils, B cells, and MDSCs. Nevertheless, the regulatory role of SOX factors in the TIME of HCC remains completely unclear. As HCC is characterized by immunologically privileged organ and immunogenic tumor, exploring its TIME and related molecular mechanisms is extremely important. Therefore, we analyzed the association between SOX factors expression and immune cells infiltrations and immune-related molecules expression, including immune checkpoint genes, antigen-presenting molecules, chemokines, and chemokine receptors in HCC by using the TIMER2 database. This section is innovative and unique in our manuscript, providing a clue that the role of SOX factors on HCC extended to affect its TIME, and paving the way for exploring the role and mechanism of SOX members in regulating TIME of HCC, and thus ultimately enabling novel immunotherapy strategies.

Previously we discussed the structure features, physiological functions, roles and molecular mechanisms of SOX family in HCC, especially in TIME. This information emphasizes the significance of SOX family in the initiation and progression of HCC. Consequently, targeting SOX family may be a potential anti-tumor therapy strategy. However, the special structure makes the SOX TFs 'undruggable' for traditional small-molecule inhibitors based on structure design. In the “SOX Transcription Factors and Translational Potential” section of our manuscript, we summarize several alternative strategies for targeting SOX with translational potential in cancer therapy and detail the experimental evidence in multiple tumors, which provides an overview of SOX-targeted therapy in cancers.

In the final section of our manuscript, "Conclusions and Outlook", we not only succinctly summarize the contents of our review, but also discuss the potential challenges of SOX family in terms of the targeted therapy in HCC, and propose some possible solutions.

Point 4: Possible application potentialities and scientific and/or technological and/or clinical impact (workflow figure?)

Response 4: Our review summarizes the clinical significance of SOX family in the diagnosis, prognosis, and therapy in HCC. For the HCC diagnosis and prognosis, some SOX factors such as SOX4 and SOX12 are significantly up-regulated in HCC tissues, and their high expression is associated with shorter overall survival times and more frequent recurrence rates. Conversely, low SOX6 or SOX7 expression predicts shorter disease-free survival and overall survival in HCC. These SOX factors are potential diagnostic and prognostic markers for HCC. For HCC therapy. SOX transcription factors are key cell fate determinants and play crucial roles in the initiation and progression of HCC. Numerous studies have demonstrated their potential as therapeutic targets in HCC. Several alternative strategies to target SOX with translational potential in cancer therapy have been described. However, there is still a considerable challenge in turning the targeted SOX factors in cancer from undruggable to reality. Our review proposes some promising SOX-targeted strategies such as PROTACs, which may contribute to the HCC therapy. Furthermore, given that TIME is critical for tumor initiation and progression and the success of immunotherapy in HCC, we explored the immune correlation of SOX factors in HCC and revealed a close association between SOX factors expression and immune cells infiltrations and immune-related molecules expression in HCC, which provided the latest clue for exploring the effect of SOX factors on TIME in HCC and developing novel combined immunotherapy strategies.

workflow figure please see the attachment.

Point 5: The authors mentioned sorafenib and appropriately mentioned ref 39, 93, 94. This reviewer personally misses some insights regarding the fact that for several years, sorafenib, has been the approved treatment option, to date, for advanced HCC patients. Its activity is the inhibition of the retrovirus-associated DNA sequences protein (RAS)/Rapidly Accelerated Fibrosarcoma protein (RAF)/mitogen-activated and extracellular-signal regulated kinase (MEK)/extracellular-signal regulated kinases (ERK) signaling pathway. However, the efficacy of sorafenib is limited by the development of drug resistance, and the major neuronal isoform of RAF, BRAF and MEK pathways play a critical and central role in HCC escape from TKIs activity. Advanced HCC patients with a BRAF mutation display a multifocal and/or more aggressive behavior with resistance to TKI (please expand referring PMID: 31766556).

Response 5: Thanks for the constructive comment. As suggested by the reviewer, we studied and cited the review (PMID: 31766556) and other literature, and proposed some of our insights regarding sorafenib resistance in the revised version. Specifically, “Of interest, blocking the Akt/mTOR signaling pathway by inhibitors significantly enhances the sensitivity of HCC cells to sorafenib, providing a potential combination therapeutic strategy for HCC [1]. Indeed, drug resistance has severely limited the efficacy of sorafenib. As a tyrosine kinase inhibitor (TKI) that mainly acts by inhibiting the RAS/RAF/MEK/ERK signaling pathway, the main neuronal isoform of RAF, BRAF, and MEK pathways and BRAF mutation are important resistance mechanisms of sorafenib [2]. Therefore, combination therapy strategies to eliminate these key resistance factors are the current priority. The activated Akt/mTOR signaling pathway has been demonstrated to confer sorafenib resistance to HCC through multiple mechanisms, including induction of Warburg shift [3] and increased TIC features [4]. The finding that SOX2-mediated the Akt/mTOR signaling pathway induces stemness and sorafenib resistance in HCC complements the stemness-related molecular mechanisms of sorafenib resistance, and further confirms the combination therapy of Akt/mTOR signaling pathway inhibitor and sorafenib for HCC is feasible.” (page 6, line 228); “As a feature of CSCs, SOX9 enhances sorafenib resistance potentially through promoting CSCs phenotypes under hypoxic conditions or activating the Akt/ABCG2 pathway [5, 6]. Of note, hypoxia, as a crucial hallmark of HCC, is responsible for sorafenib resistance to HCC. The mechanisms underlying hypoxia-induced HCC cells escape from sorafenib include HIF-mediated metabolic reprogramming [7], regulation of the PI3K/AKT signaling pathway [8], etc. The above-mentioned study on SOX9 further expands the mechanism of hypoxia-induced sorafenib resistance and closely links hypoxia, CSCs, and sorafenib sensitivity.” (page 9, line 392).

  1. Wen W, Han T, Chen C, Huang L, Sun W, Wang X et al. Cyclin G1 expands liver tumor-initiating cells by Sox2 induction via Akt/mTOR signaling. Mol Cancer Ther. 2013; 12: 1796-1804.
  2. Gnoni A, Licchetta A, Memeo R, Argentiero A, Solimando AG, Longo V et al. Role of BRAF in Hepatocellular Carcinoma: A Rationale for Future Targeted Cancer Therapies. Medicina (Kaunas). 2019; 55.
  3. Krstic J, Reinisch I, Schindlmaier K, Galhuber M, Riahi Z, Berger N et al. Fasting improves therapeutic response in hepatocellular carcinoma through p53-dependent metabolic synergism. Sci Adv. 2022; 8: eabh2635.
  4. Loh JJ, Li TW, Zhou L, Wong TL, Liu X, Ma VWS et al. FSTL1 Secreted by Activated Fibroblasts Promotes Hepatocellular Carcinoma Metastasis and Stemness. Cancer Res. 2021; 81: 5692-5705.
  5. Xiao Y, Sun Y, Liu G, Zhao J, Gao Y, Yeh S et al. Androgen receptor (AR)/miR-520f-3p/SOX9 signaling is involved in altering hepatocellular carcinoma (HCC) cell sensitivity to the Sorafenib therapy under hypoxia via increasing cancer stem cells phenotype. Cancer Lett. 2019; 444: 175-187.
  6. Wang M, Wang Z, Zhi X, Ding W, Xiong J, Tao T et al. SOX9 enhances sorafenib resistance through upregulating ABCG2 expression in hepatocellular carcinoma. Biomed Pharmacother. 2020; 129: 110315.
  7. Bao MH, Wong CC. Hypoxia, Metabolic Reprogramming, and Drug Resistance in Liver Cancer. Cells. 2021; 10.
  8. Zeng Z, Lu Q, Liu Y, Zhao J, Zhang Q, Hu L et al. Effect of the Hypoxia Inducible Factor on Sorafenib Resistance of Hepatocellular Carcinoma. Front Oncol. 2021; 11: 641522.

Round 2

Reviewer 2 Report

The authors addressed the major points.

Reviewer 3 Report

The authors have clarified several of the questions I raised in my previous review. Most of the major problems have been addressed by this revision.